# Balancing fire risk and human thermal comfort in fire-prone urban landscapes

Tania A. MacLeod[1,2¤a]*, Amy K. Hahs[2¤b], Trent D. Penman[1]

**1** School of Ecosystem and Forest Sciences, Bushfire Behaviour and Management, The University of Melbourne, Creswick, Victoria, Australia, **2** Royal Botanic Gardens Victoria, Australian Research Centre for Urban Ecology (ARCUE), c/o School of BioSciences, The University of Melbourne, Parkville, Victoria, Australia

¤a Current address: City of Greater Bendigo, Public Space Design Unit, Bendigo, Victoria, Australia
¤b Current address: School of Ecosystem and Forest Sciences, Green Infrastructure Research Group, The University of Melbourne, Richmond, Victoria, Australia
* tania_macleod@yahoo.com.au

**Data Availability Statement:** All relevant data are within the paper and its Supporting Information files.

**Funding:** The authors received no specific funding for this work.

## Abstract

Vegetation in urban areas provides many essential ecosystem services. These services may be indirect, such as carbon sequestration and biological diversity, or direct, including microclimate regulation and cultural values. As the global population is becoming ever more urbanized these services will be increasingly vital to the quality of life in urban areas. Due to the combined effects of shading and evapotranspiration, trees have the potential to cool urban microclimates and mitigate urban heat, reduce thermal discomfort and help to create comfortable outdoor spaces for people. Understory vegetation in the form of shrubs and grass layers are also increasingly recognized for the positive role they play in human aesthetics and supporting biodiversity. However, in fire-prone urban landscapes there are risks associated with having denser and more complex vegetation in public open spaces. We investigated the effects of plant selection and planting arrangement on fire risk and human thermal comfort using the Forest Flammability Model and Physiological Equivalent Temperature (PET), to identify how planting arrangement can help balance the trade-offs between these risks and benefits. Our research demonstrated the importance of vertical separation of height strata and suggests that Clumped and Continuous planting arrangements are the most effective way of keeping complex vegetation in public open space to deliver the greatest human thermal comfort benefit while minimizing potential fire behaviour. This study provides an example of how existing research tools in multiple ecological fields can be combined to inform positive outcomes for people and nature in urban landscapes.

## Introduction

The 2014 Intergovernmental Panel on Climate Change (IPCC) report on climate projections for south-eastern Australia predicts that increased warming will be associated with not only more frequent and extreme fire weather but more frequent heat waves [1]. The IPCC warns of links between climate change and human health including increased rates of disease and morbidity [2], as well as the increasing vulnerability of many ecosystems [1]. These effects are

**Competing interests:** The authors have declared that no competing interests exist.

amplified in urban areas [2] where the prevalence of paved surfaces leads to increased solar absorption, creating local environments that that are often several degrees warmer than surrounding areas, further intensifying heat events with an often deleterious effect on human health [3,4].

It is widely accepted [5–7] that vegetation in urban areas provides many essential ecosystem services. These services may be indirect, such as carbon sequestration and biological diversity, or direct, including microclimate regulation and cultural values. As such, Urban Forest programs within cities are proliferating, as the importance of vegetation in human settlements is increasingly recognised. Due to the combined effects of shading and evapotranspiration, trees have the potential to cool urban microclimates thus help mitigate urban heat, reduce thermal discomfort and help create comfortable outdoor spaces for people [3, 4, 8]. As the global population is becoming ever more urbanised these services will be increasingly vital to the quality of life in urban areas.

However, increased vegetation within the urban landscape has its own risks, particularly where communities are extremely vulnerable to the effects of bushfire [9]. The proximity of native vegetation to human settlements, the accumulation of high fuel loads and increased human activity has led to increased incidents of ignition [9–11], often resulting in significant loss of life and property and profound economic and environmental damage [12]. Although the threat from fire cannot be totally eradicated from the urban interface, several methods have been adopted to reduce the risk of fire to human settlements including land-use planning provisions, building controls and fuel reduction activities [13–15] that frequently result in the indiscriminent clearing of trees and vegetation.

Melbourne's urban interface is located in one of the most fire prone regions on earth [16]. At the same time, Victoria's population is increasing at the second-fastest rate of all Australian states and territories with Melbourne experiencing the largest growth of all capital cities between June 2013 and June 2014. Greater Melbourne accounted for ninety per cent of Victoria's growth in the same period with many areas on Melbourne's urban growth boundary experiencing the largest growth of all areas in Australia [17]. The population of Greater Melbourne is projected to increase continuously to as many as 9.8 million by 2061 [17].

These dual risks of increased temperatures and altered fire patterns are poignantly illustrated by the climatic events associated with the devastating bushfires of Black Saturday, 7 February 2009. While the bushfires themselves resulted in the death of 173 Victorians [10], the heat wave leading up to the events of Black Saturday included three days above 43˚C, which resulted in 374 excess deaths compared to the same period the previous year [18]. It has been reported that that extreme heat is responsible for more deaths than any other natural disaster in Australia [19]. The heatwave of 2009 highlights the serious nature of heat as a killer and the need to begin identifying how to design and manage urban landscapes that support nature-based solutions that help reduce high temperatures experienced by human populations, yet minimize any additional risks related to bushfire spread and severity.

This study sought to examine how plant selection, vegetation structure and planting arrangements affect the often-competing values of community bushfire safety and urban heat mitigation and offers recommendations for the planning and management of public open space in Melbourne's urban interface.

## Methods

### Geographic context for study

The study was conducted based on typical climatic and biogeographic conditions found in south-eastern Australia, with many of the field measurements taken in locations in and around

**Table 1. Climatic and spatial data used in the study of fire behaviour and urban heat for open space in Greater Melbourne.**

| Input | Measurement | Description and Source |
|---|---|---|
| Air Temperature | 31.78˚C | Taken from the 95[th] percentile of summer (O, N, D, J, F, M) maximum daily temperatures. This is the average of 8 weather stations located in Melbourne's outer fringe [20] |
| Relative humidity | 48.8% | Mean Summer (Oct, Nov, Dec, Jan Feb Mar) 3pm relative humidity (%) for years 1994 to 2010 from 7 weather stations located in Melbourne's outer fringe [20] |
| Wind speed | 19.6km h$^{-1}$ (5.44 m s$^{-1}$) | Mean Summer (Oct, Nov, Dec, Jan Feb Mar) 3pm wind speed (km/h) for years 1994 to 2010. This is the average of 7 weather stations located in Melbourne's outer fringe [19] |
| Altitude | 216m | Taken from digital elevation shapefile [21]. This is a neighbourhood park location in Craigieburn. |
| Vapour Pressure | 22.9 (hPa) | Calculated in RayMan Pro from relative humidity |
| Cloud Cover | 0.0 | 0% cloud cover |

Greater Melbourne (Table 1). Melbourne is located at 144˚0' longitude and -37˚0' latitude with a Mediterranean climate, consisting of hot dry summers, and cooler winters. For the purpose of this study, we quantified the climatic conditions at 3pm for a 95[th] percentile summer day, where conditions will be warmer than a usual summer day, without representing an extreme heat event.

## Planting arrangement

Planting Arrangements vary in 1) the plant species and their associated morphology (e.g. canopy height, leaf area, deciduous/evergreen); 2) the spatial arrangement of the plants across the horizontal surface of a space; and 3) the vertical profile of the planting (ie., the presence of a shrub layer and/or an herbaceous understory. For this study we used a multi-factorial design to test multiple combinations of planting design elements within a hypothetical square park with an area of 1 ha, and an average surface fuel load of 8 tonnes per ha, which is roughly equivalent to a thin layer of mulch or leaves (Fig 1). In the initial study we also tested slope and a higher surface fuel load however increasing the slope from flat to 5 degrees did not have a discernible impact on the rate of spread of the fire. Increasing the fuel load from 8 to 15 tonnes per hectare only slightly increased the height of the flame.

We conducted an initial field and desktop assessment of public open spaces around Melbourne's northern and eastern suburbs to determine 1) frequently observed plant species, 2), tree canopy density 3) spacing between trees and other vegetation and 4) average fuel surface loads (volume of mulch or other flammable ground surface materials) to ensure our planting arrangements represented realistic scenarios). The public open spaces observed ranged from more intensively managed parks and gardens to natural bushland reserves. For the most part, these were unirrigated landscapes with depleted topsoil and very little natural or applied near surface fuels. Public open spaces visited were municipal land managed by the Darebin, Yarra, Whitehorse City and Mitchell Shire Councils (Table A in S2 File). No permission was required for the site observations given all sites were public spaces and all desktop observations were undertaken using GIS and open source spatial files [21]. We used this initial assessment to refine the following parameters to represent commonly observed planting design elements.

Tree forms:

• Large native tree species, open spreading canopy (e.g. *Eucalyptus camaldulensis*)

• Medium sized, introduced deciduous tree (e.g. *Quercus robur*)

• Medium native tree species, denser more compact canopy (e.g. *Eucalyptus sideroxylon*)

## Canopy Trees

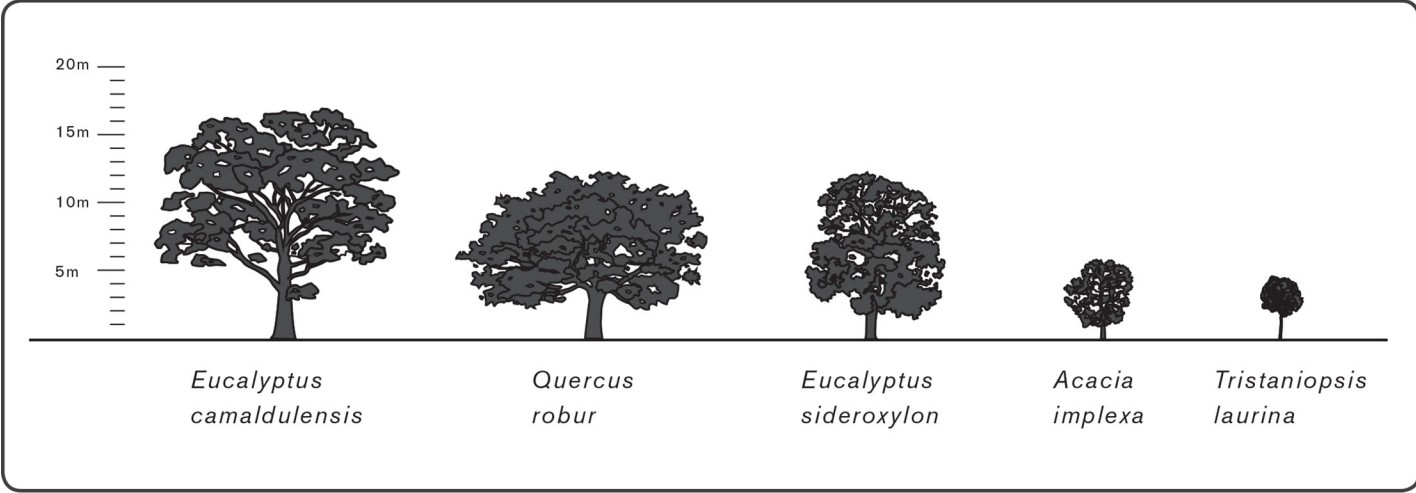

## Planting Arrangement

**X**

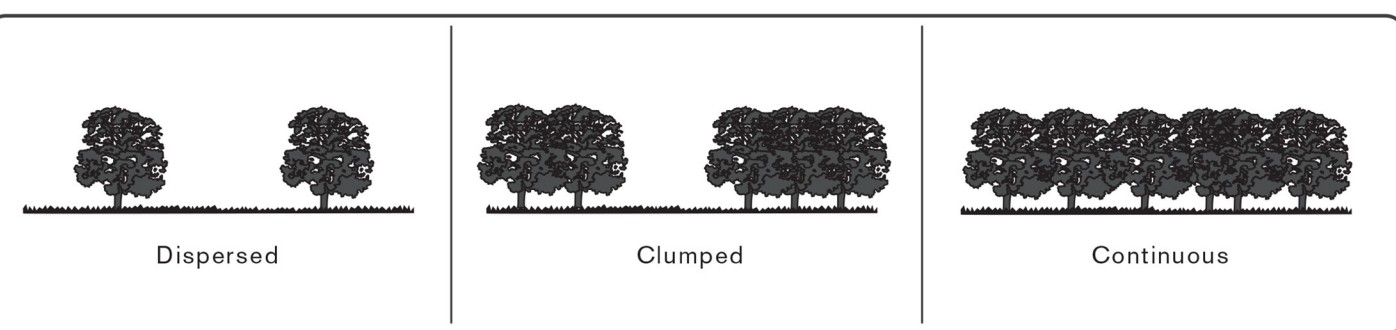

## Understorey Vegetation

**X**

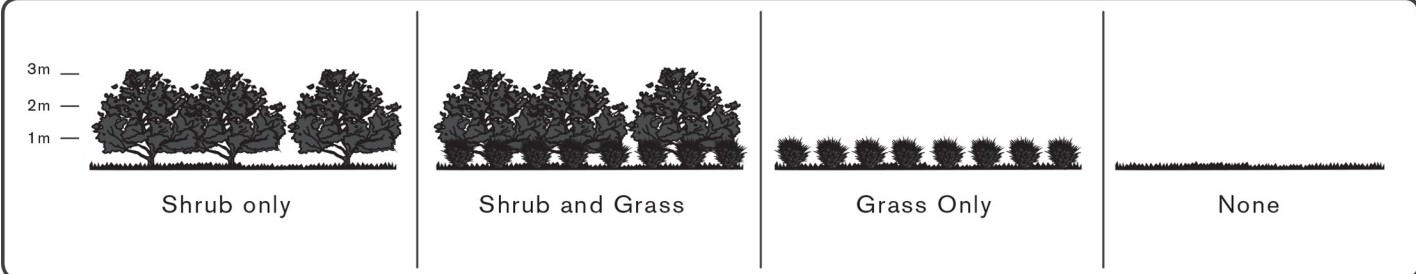

**Fig 1. Three key planting design elements, and their variations, used to create the planting arrangements tested in this project.**

- Small sized native tree species, open canopy (e.g. *Acacia implexa*)

- Small sized native tree species, denser more compact canopy (e.g. *Tristaniopsis laurina*)

Understorey species were represented by frequently observed understorey plants, or elevated and near surface fuels, during our initial field assessments. One shrub and one grass

were selected—*Bursaria spinosa*, a medium sized, open canopy native shrub, and *Poa labillardieri* a native tussock grass.

## Evaluating effect of planting arrangement on flammability

The Forest Flammability Model (FFM) developed by Zylstra [22] was used to examine the effects of vegetation arrangement on fire behaviour. Other fire models predict fire behaviour from fuel load or fuel hazard. In contrast, the FFM recognises the importance of species traits and models the propagation of flames through vegetation by the characteristics of each species (e.g. leaf shape and arrangement, ignition potential) and the transfer of heat between them. Climate is included in the model through temperature and wind speed. We selected the 95[th] percentile values of seven weather stations across Melbourne (31.78 °C, 19.6 km h$^{-1}$).

Flammability of each individual planting combination was measured by flame height and rate of spread. Flame height is determined by the amount of vegetation that is available as fuel to burn and influences the likelihood of successful suppression attempts [22]. A safe distance is generally four-times the maximum flame height [23] while there is little if any chance of suppressing flames exceeding four meters [24,25]. Rate of spread is influenced by the fuel structure, weather and topography. Faster spreading fires have greater potential for damage as they are difficult to contain.

Plant trait information for each of the modelled plant species were collected from published sources, unpublished studies, the online TRY plant database [26, 27] and from herbarium samples (Tables A and B in S1 File). Additional field collected data was required to measure plant traits that could not be sourced from existing information. Plant dimensions were taken from an average of field collected measurements for each of the species studied. Dimensions were canopy base height, crown height, crown width, crown base edge height and crown top edge height and were collected from plants that were a good representation of a mature specimen of that species growing in Melbourne.

The C++ code for the FFM was accessed freely from https://github.com/pzylstra/ffm_cpp and run in R-studio. To compare the mean and standard deviation of flame height and rate of spread we ran 1,000 Monte Carlo iterations of the FFM for each Planting Design scenario. Uncertainty was captured throughout the variation in the input data required to run the model. A description of the specific model inputs is listed in Appendix I.

## Evaluating effect of planting arrangements on human thermal comfort

We calculated the average physical equivalent temperature (PET) for each Planting Design scenario to investigate how the Planting Design scenarios influenced human thermal comfort. PET is defined as the equivalent air temperature at which, in a typical indoor setting without solar radiation and wind, the heat budget of the human body is balanced with the conditions being assessed [28]. Temperatures of around 20°C are considered comfortable while the body starts to experience heat stress above 23°C for an average fit young male with an average metabolism of 80–100 W of light activity and the heat resistance of light clothing, [28–30].

PET was calculated using the RayMan Pro Model developed by Matzarakis, Rutz [31] The RayMan Pro Model simulates long- and short-wave radiation in the environment in order to calculate mean radiant temperature ($T_{mrt}$) without the need to directly measure radiation fluxes using input data consisting of hemispherical imagery, climatic data, date and time information, geographic data and personal data. $T_{mrt}$ is described as the uniform temperature of an imaginary black enclosure in which a person would have the same energy balance as the surrounding environment [32]. By including time, location and cloud cover data into the model, values for global radiation were simulated to enable to $T_{mrt}$ and PET to be calculated [33, 34].

Inputs for the RayMan model include hemispherical imagery, climatic data, date and time information, geographic data and personal data. Climatic data matched the inputs to FFM with temperature (31.78°C) and wind speed (19.6 km h$^{-1}$), and the additional factors of relative humidity (48.8%) and cloud cover. Date and time were set as the 31$^{st}$ January at 3pm to represent a hot summers day. Geographic data were set for the centre of Melbourne (as described above) with an elevation of 216m above sea level. The output is PET which can be interpreted into human thermal comfort following the categories described by Fröhlich and Matzarakis [33].

An advantage of the RayMan model is that it requires only basic meteorological data and a limited number of other inputs to calculate mean radiant temperature. It is also a very widely used tool, validated by many previous studies, is quick and easy to use and freely available from its authors [35, 36].

The effect of tree canopy species and planting arrangement on human thermal comfort in this study was modelled in a similar way to that employed by Fröhlich and Matzarakis [33] to measure changes in street tree design and surface material on human thermal comfort in Freiburg, Germany. They imported hemispherical imagery into Rayman in order to calculate sky view factor (SVF) and PET to analyse changes in human thermal comfort as a result of changes in the distribution of vegetation.

SVF is a measure ranging from 0 and 1 and represents the portion of sky that is obstructed [34] and therefore the amount of shade. Highly shaded conditions have a SVF of less than 0.3 and generally experience cooler summer temperatures than less shaded areas [37]. Hemispherical imagery of each tree species and planting arrangement was imported into RayMan Pro (Tables A and B in S2 File) and the model was run to produce an average SVF and PET value for each.

Since the SVF is captured above the level of the understory vegetation, we calculated human thermal comfort for a reduced set of planting arrangement scenarios. Unlike Fröhlich and Matzarakis [33], radiation and shade from surrounding built structures and surfaces was not studied–only the effect of each tree species in each planting arrangement. The parameters used as input data for the RayMan Pro Models can be found in appendix 2.

### Managing the trade-offs through planting arrangement

To investigate how planting design could be used to manage the trade-offs between fire spread, fire severity and human thermal comfort, we compared Flame height and Rate of Spread values for each scenario with the Physiological Equivalent Temperature of the associated canopy configuration. Trends in the data are visually identified and discussed.

The inputs, variables and measures used for both the FFM and RayMan Pro models are shown in Table 2.

## Results

### Flammability

The presence of *B. spinosa* in the shrub layer greatly increased Flame height, both with and without a grassy understorey, with all scenarios producing Flame heights of more than 4m, and maximum Flame heights equivalent to the height of the tree species (Fig 2). Sites with only *P. labillardieri* in the understory had lower Flame Heights (generally < 4m), and the most rapid Rate of Spread (Fig 2). The sites with an absence of understory had the lowest Flame heights and slowest Rates of Spread (Fig 2).

However, there were variations within this larger trend depending upon the morphology of the canopy species and the planting arrangements (Fig 3). The maximum Flame height was

**Table 2. The models and their respective inputs for the study of flammability and human thermal comfort of the open space scenarios.**

|  | Flammability | Human thermal comfort |
|---|---|---|
| **Model** | Forest Flammability Model (FFM) [21] | RayMan Pro [31] |
| **Inputs** | Climatic data<br>Plant trait data<br>Plant spacing<br>Slope<br>Presence or absence of elevated, near surface and surface fuels | Climatic data<br>Hemispherical photographs<br>Personal data<br>Geographical data |
| **Measure** | Flame height<br>Rate of spread | Sky view factor (SVF)<br>Physiological Equivalent Temperature (PET) |
| **Variables** | • Planting arrangement<br>• Canopy species<br>• Elevated and near surface fuels<br>• Surface fuel load<br>• Slope | • Planting arrangement<br>• Canopy species |

relative small for the two small sized trees (*Acacia implexa* and *Tristaniopsis laurina*; ~ 6m and 9 m respectively; Fig 3A and 3E), compared to the larger trees (E. *camaldulensis* and *E. sideroxylon*; 14 m and 13 m respectively; Fig 3B and 3C); while the maximum Flame height was

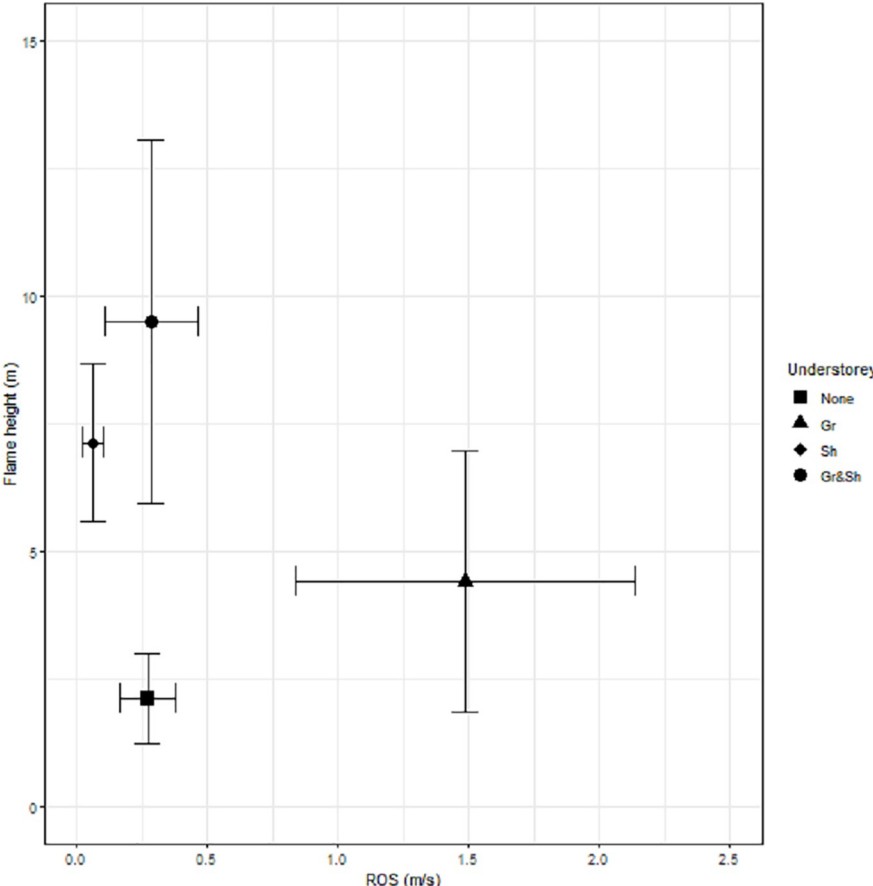

**Fig 2. Flame height and rate of spread for the four different combinations of understorey.** The data show the mean and standard error for the full combination of canopy species, planting arrangements, mulch depths and slopes.

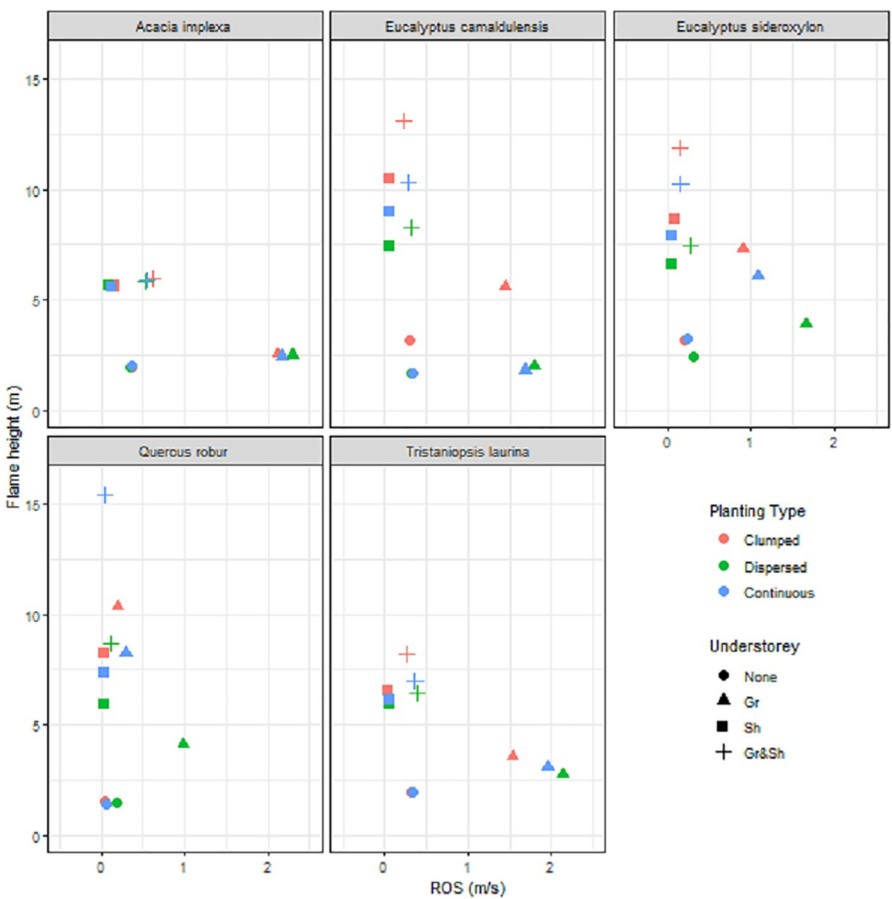

**Fig 3. Flame height and rate of spread for each of the five canopy species.** The symbols represent the four different combinations of understorey, while the colours represent the different planting arrangements. Each point represents the mean value for the four mulch depth and slope combinations.

highest for the medium deciduous tree (*Quercus robur* Fig 3D). Planting arrangements where elevated and near surface fuels were both present consistently led to relatively larger Flame heights and more rapid Rates of Spread compared to scenarios where they were absent. However, the spatial arrangement of the plantings showed consistent impacts on the two flammability measures, with clumped plantings leading to slower Rates of Spread (e.g. Fig 3E) but potentially higher Flame heights (e.g. Fig 3B); while Dispersed Plantings consistently lead to the most rapid Rates of Spread and lower Flame heights (e.g. Fig 3C and 3E). Continuous planting arrangements displayed a moderate response between the other two planting arrangements.

## Human thermal comfort

The Physiological Equivalent Temperature showed a 1.8˚C variation across the fifteen Planting arrangement scenarios, covering an almost complete range of values between a PET of 38.9˚C under no canopy cover to a PET of 35.1˚C under complete canopy cover (Fig 4). The trees species with the denser canopies (e.g. *Q. robur*, *T. laurina*, *E. sideroxylon*) provided the greatest cooling benefit; as did the Continuous followed by the Clumped planting arrangements. The tree species with the least impact on PET were those with a more open canopy (e.g., *E. camaldulensis*, *A. implexa*); particularly when they were planted in a Dispersed spatial arrangement.

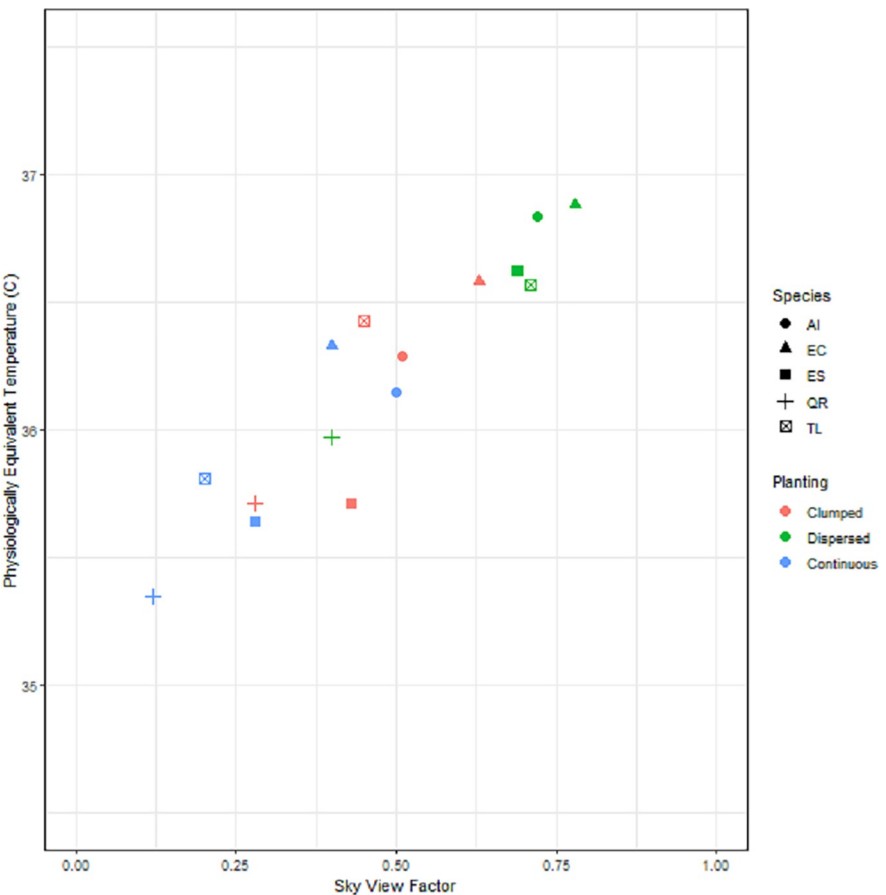

**Fig 4. The relationship between sky view factor (SVF) and physiological equivalent temperature (PET).**
PET = 35.166 + 2.167(SVF). Each point represents the mean SVF and PET values for each for each canopy species in each planting arrangement taken from multiple hemispherical images and corresponding RayMan outputs. AI = *Acacia implexa*, EC = *Eucalyptus camaldulensis*, ES = *Eucalyptus sideroxylon*, QR = *Quercus robur*, TL = *Tristaniopsis laurina*.

Trade-offs between Flammability and Human Thermal ComfortThe planting arrangements that led to the coolest PET and lowest Rates of Spread for fire (Fig 5) were those with a Continuous or Clumped arrangement, and medium sized denser tree canopies (e.g. *Q. robur*, *E. sideroxylon*). Even when the grassy understory was present, the Rate of Spread for fires was slower than that observed in the remaining planting arrangement scenarios. However, these two canopy species showed different responses in terms of Flame Height for the different scenarios (Fig 5). Flame Heights for *E. sideroxylon* displayed less variability and a shorter mean Flame height compared to those of *Q. robur*, which showed much greater variability depending upon the nature of the understorey vegetation.

## Discussion

### Effects of planting arrangement on flammability

Our results demonstrate that the inclusion of a *B. spinosa* shrub layer beneath canopy trees has a marked impact on the flame height regardless of the arrangement of the tree species (Fig 2). The highest flames were produced by the taller tree species in the presence of understorey (Fig 3) explained by the ladder effect in which understorey fuels are able to carry and sustain the

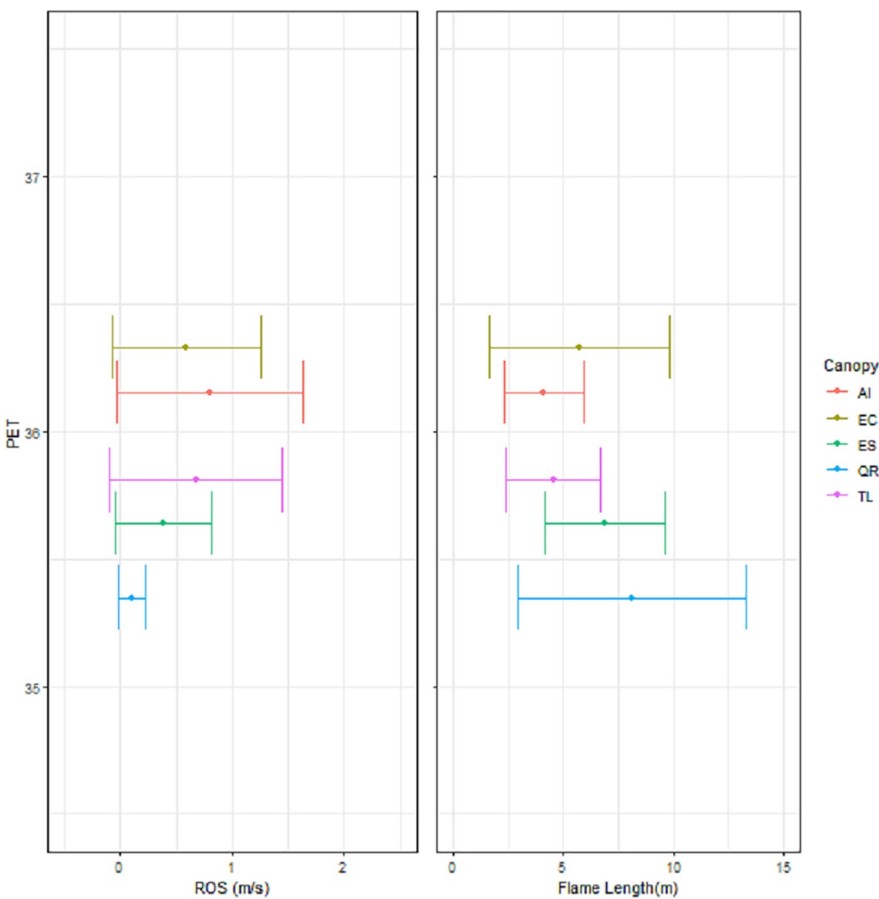

**Fig 5. Comparison of rate of spread (m/s) and flame height (m) for the physical equivalent temperature (PET) of each canopy species under the continuous planting arrangment.** AI = *Acacia implexa*, EC = *Eucalyptus camaldulensis*, ES = *Eucalyptus sideroxylon*, QR = *Quercus robur*, TL = *Tristaniopsis laurina*.

flame from ground level up into the tree canopy to produce a much more locally intense fire [38].

Given that the average height of the shrub species tested (*B. spinosa*) was 3.3 meters and the foliage of the canopy only ten centimetres off the ground, the presence of a shrub layer in this case overlapped with fuel elements from the other strata providing the vertical continuity needed for the fire to consume the entire stand in the majority of scenarios tested. *B.spinosa* is an indigenous shrub that is naturally occurring and regularly planted in these landscapes. It forms a relatively tall shrub, with an architecture that is conducive to higher flame heights [39]. Shrub species with alternative plant architectures will produce a different flammability response.

The only two planting arrangements with a shrub layer in which the flames did not engulf the entire tree canopy were those with *E. camaldulensis* planted in Dispersed and Continuous arrangements (Fig 4B). In both of these arrangements the base heights of the tree canopies were well above the top of shrub layer providing vertical separation between fuel elements, providing further insights into management actions that can contribute to reduced fire risk in urban landscapes.

The addition of a grassy understory contributed further to slightly greater flame heights and slightly faster rates of spread compared to the presence of a shrub layer alone. Therefore,

in planting arrangements where a tall shrub understory is desired, limiting the addition of a grassy understory may help reduce the intensity of fire behaviour in these systems and therefore increase the likelihood of successful suppression.

Modes of fire spread were also influenced by the arrangement of the overstorey trees. Although scenarios with Clumped groups of trees and a shrub understorey resulted in the greatest flame heights, they also had the greatest impact on supressing wind speed, acting as a wind break and slowing the spread of the flames. In scenarios with a *P. labillardieri* understory, the Dispersed planting arrangement of trees led to the fastest Rates of Spread, followed by the Continuous, then Clumped arrangements, regardless of the overstory tree species (Fig 3). This reflects more rapid spread of fires in grassy systems.

The results suggests that where the aim is to include a grassy understory, a clumped arrangement of tall trees without a shrub layer or overlapping fuel strata is likely to provide the lowest flame heights and slowest rates of spread under the modelled conditions as there are greater distances between ground level fuel elements, and greater wind interference due to the clumped arrangement of trees. However, for planting designs where a shrub layer is desired, plantings of trees in a Dispersed arrangement will help reduce Flame height, without greatly influencing the Rate of Spread, as the fire will take more time to consume individual shrubs compared to clumps of shrubs that offer a higher localised fuel load. A reduced flame height will also help reduce the impact of the fire on the tree canopy, as demonstrated by the *E. camaldulensis*, where the flame height for the Dispersed arrangement was roughly half the height of the tree canopy, and five meters lower than the flame height in the Continuous scenario (Fig 3C).

The results suggest that patches of multi-strata vegetation may be favourable to continuous plantings by breaking up the continuity of fuels, slowing fire spread and allowing for fire breaks to be created. From the hypothetic scenarios tested, provided there is enough separation between clumps of trees and shrubs, Clumped planting arrangements slow the spread of fires, thereby buying time and aiding suppressing attempts [23]. Such patches could also allow for localised cooler environments or refuges and for understory to continue to be an important feature of urban ecosystems.

Our modelled results show that planting arrangements with the greatest vertical and horizontal separation between trees in the presence of a grassy layer produced the fastest moving, yet the shortest flames. For urban landscapes, where green spaces are often bordered by a local road on at least one of their boundaries, Flame heights of up to 1.5–2.0 m are unlikely to travel into the surrounding landscape. Flames that are less than two meters can also often easily be fought by ground personnel [24, 25] and are associated with a lower risk of asset damage. Therefore, for most of the planting scenarios we tested, even those with a *P. labillardieri* only understory, the fires are unlikely to spread across the width of a local road (Fig 3) under the modelled climatic conditions and will be easier to prevent damage to property. The potential exceptions are when small trees form the canopy layer, as these smaller tree species are unlikely to provide the same level of wind dampening at ground level as larger trees that create greater interruption to the local wind flow. Therefore, limiting the amount of grassy understory below these shorter trees could help reduce the likelihood of scenarios with a more rapid Rate of Spread for fire. Placing taller trees further away from the edges of urban greenspaces may also help to reduce the risk of fire travelling into the adjacent landscape.

## Effects of planting arrangement on human thermal comfort

The greatest PET benefits were obtained from Clumped followed by Continuous planting arrangements, and species with denser tree canopies (*Q. robur*, *E. sideroxylon*, *T. laurina*).

This information can be used to inform planting designs when a key desired outcome is to provide an open space that offers human visitors a reduced exposure to high temperatures and solar radiation. Interestingly, these arrangements are often observed in existing public open spaces, with Continuous scenarios equivalent to avenues that are often planted along footpaths or boundaries within parks, and the Clumped scenario reflecting a common European park-land style layout. Our research offers the additional benefit of providing evidence-based guidance around how more complex understory assemblages can be structured to support biodiversity and aesthetic outcomes within these planting arrangements without greatly impacting upon fire risk in those landscapes. The most promising results for both slowing fire spread and improving human comfort were achieved in scenarios with Continuous or Clumped arrangements of large trees with denser foliage cover, namely plantings of *Q. robur* in the absence of understorey fuels. The dense canopy cover provided by *Q. robur* appears to be having a dampening effect on the wind on the flame which is resulting in a more localised intense but slower moving fire. Large tree species also have the added benefit of trapping embers in their canopies thereby further minimising fire spread. This supports advice given by the CFA [40] that well positioned large trees can help in fire suppression efforts by reducing the localised wind speed and the spread of the fire. Trees can also shield other potential sources of ignition such as embers and radiant heat emanating from the fire line [23]. The result is not only slower moving, ground level fires but a greater amount of shade and thermal comfort. The superior shade coverage produced by *Q. robur* compared to the more open Eucalypt canopies and smaller trees species resulted in a lowering of PET thereby improving human comfort.

It should be noted that the reduction in PET in this study is a result of the sky view factor or per centage of shade produced by the tree canopy in each planting scenario alone therefore the demonstrated reduction in PET beneath tree canopies is attributed to the shielding of solar radiation by tree branches, trunks and foliage. As a result of this shading, below-canopy soil and air temperatures may be substantially cooler from comparable sparser areas that receive greater solar radiation [41]. This shielding also reduces wind speed and the mixing of air beneath canopies resulting in higher relative humidity and evapotranspiration. As such, humidity under tree canopies generally decreases as the canopy becomes more open [41, 42]. The climate inputs including relative humidity and windspeed into the RayMan model remained equal for all planting scenarios in this study. Results would be further strengthened by a more accurate prediction of the effect of relative humidity and windspeed on evapotranspiration and human thermal comfort for each of the planting scenarios tested. Whilst shorter tree species may produce shorter fires, their height and size relative to the human scale is an important consideration when planting trees for effective shade and heat mitigation. In order to realise the cooling benefits from a continuous planting of *T. laurina* for example, considerably more trees would be required to get the same cooling effect over the same area given their relatively small size and canopy dimensions.

## Implications for planning and management

In response to recommendations from the 2009 Victorian Bushfires Royal Commission the Victorian Planning Provisions (VPP) were amended (VC83) to strengthen community resilience to bushfire. Together with the introduction of a Bushfire Management Overlay (BMO) the primary purpose of these changes was to prioritise human life over other planning criteria [15–17]. In order to create defendable space around human settlements, the new provision enables permit exemptions for the removal of any vegetation within 10m and trees within 50m of existing dwellings for areas covered by a BMO. For the rest of the State, except in what

could be considered established built-up local government areas, permits are not required to remove vegetation within 10m or trees within 30m of an existing building [15, 17]. Yet blanket planning policies and overlays such as the 10/30 and 10/50 exemptions in Victoria and NSW for the clearing of all trees and vegetation in proximity to human settlements fail to address the complexities and traits of the vegetation itself and the role that vegetation plays in manipulating fire behaviour but also in making urban areas more liveable.

With respect to minimising fire spread and severity, our results support local planning policies and advice given from fire agencies that seek to minimise the presence of tall trees and continuous fuels around properties to aid in firefighting and asset protection efforts. Yet in recognition of the important role these vegetation elements play in supporting biodiversity within urban landscapes, our research also indicates that there are design and management approaches available that can maximise the benefits, while reducing the potential fire risk through design.

Recognising that people who live in the urban-interface often do so due to lifestyle and amenity preferences, there have been several efforts both locally [40, 43–46] and abroad [47] to better inform constituents of 'firewise' gardening. Although these documents acknowledge the importance of vegetation in residential areas for other values, their focus is primarily on asset protection rather than ecosystem service. Landscape management suggestions to minimise fire threat in residential areas include the removal of vegetation beneath trees, pruning shrubs away from tree branches, the removal of dead plant material and using gravel paths and lawn to break up separation between fire hazards. Our results would indicate that all are sound advice if reducing fire threat to human life and settlements are the primary objectives. Other listed management interventions to reduce fire intensity include inorganic mulches, compacting surface fuels and materials more tightly and introducing lush, green understorey species to increase the relative humidity of the microclimate [46, 48].There may be opportunities for open space managers to manipulate and modify vegetation to reduce fire risk within planting arrangements. The results of this study support recommendations that uplifting canopies and creating greater vertical separation between the individual fuel strata could be a management option for influencing fire behaviour in public open space [48].

Just as there are differences in flammability between plant species and compositions, there are differences in flammability within species due to plant traits. This study and further application of the FFM in managed urban landscapes would be further strengthened by factoring in the differences of plant traits of individual species under each of the planting arrangements and a greater consideration of fuel moisture content. Studies by Murray, Hardstaff [49] and Krix and Murray [50] found variation in leaf flammability of the same species between fresh and dry leaves and across landscape gradients respectively. In both studies as well as several others [51, 52] leaf moisture content appears to be a key determinant of plant flammability by creating an important buffering effect and slowing the time to ignition yet was not tested in this study Landscape managers may choose to reduce the flammability of vegetation in urban areas by increasing the amount of moisture that is applied and retained in the landscape, especially over the summer months. Moist soils correspond to an increase in evapotranspiration and cooler near-surface air temperatures [41] and determine how much water is available for uptake by plants. Increasing water availability to vegetation may reduce and the extent to which vegetation dries out, reducing its flammability and increasing the cooling effect of the vegetation through evapotranspiration.

Of course, all management decisions will have resource implications. Water security in the study area will become increasingly pressing over the coming years. More intensive management options may achievable in smaller gardens and high-profile civic spaces yet be totally impractical for larger parks, natural bushland areas and expanses of open space.

## Limitations and future research

The results from this study would be greatly strengthened through the direct measurement of the microclimate and plant trait data from real examples of the different open space scenarios. Given time limitations, the plant trait data for the FFM was largely collected from previously studies, published literature and from international data-bases and therefore was not specific to the study area. It was also assumed in the flammability modelling that all of the vegetation was alive. The collection of plant trait data at different times of the year could provide an accurate indication of how seasonal climate and rainfall impacts fire behaviour in the different planting scenarios. It would also enable analysis of the impact of management interventions such as summer irrigation on landscape flammability. Field measurements of the microclimate of planting scenarios would also allow for the effect of understorey vegetation and evapotranspiration on human thermal comfort to be analysed.

Whilst continuous plantings of large deciduous trees with no understorey resulted in cooler temperatures and a reduced fire risk under the hypothetical scenarios presented, extensive plantings of such arrangements will have trade-offs in other ecosystem services such as biodiversity, the conservation of native plant and animal species and soil health. The omission of understorey and flammable surface fuels from urban areas could have deleterious effects for many bird species that rely on the complexity of understorey for shelter, foraging and nesting. A study by Threlfall, Williams [53] suggests that increasing the diversity and quantity of understorey, leaf litter and coarse woody debris in urban areas can benefit both bird and bat assemblages. The presence of leaf litter and debris also forms important habitat for many ant species [54]. Recommended methods to aid the recruitment of native plant species such as retaining fallen deadwood, leaf litter and mulch [55, 56] could be considered dangerous in public open space in bushfire prone areas, although our findings suggests that there may be planting arrangements that can help reduce the associated fire risks.

We do not suggest that we need to omit certain plant species and combinations from human settlements altogether but we need to be mindful of how we plan new developments and manage landscapes in fire-prone urban areas. For example, planting arrangements in this study with an *E. camaldulensis* overstorey performed poorly in respect to both heat mitigation and fire risk. At face value one could conclude that *E. camaldulensis* is an unsuitable tree species where both thermal comfort and bushfire risk reduction are a priority. Yet these landscapes are typical of many open spaces in Melbourne's expanding suburbs and have important cultural [57] and ecological values [56]. An understanding of the traits of this species may help determine the best design approach and provide insights into how we can best integrate *E. camaldulensis* trees in the urban landscape.

This study was limited by the fact that human thermal comfort and fire threat was analysed from hypothetical scenarios of a restricted number of plant species under one set of climatic conditions. Results would be strengthened from studying the effects of a greater number of plant species; especially the effects of understorey on flame height as only one elevated fuel and one near surface fuel were tested. Flammability in scenarios with elevated fuels were largely influenced by the dimensions and spacing of *B. spinosa*. This shrub species not only overlapped vertically with canopy and surface fuels; individual plants of this species overlapped horizontally as they were often densely planted. Testing the effects of different spacing between individual understorey plants on landscape flammability is recommended.

The study did not consider how the planting arrangements performed with climatic conditions predicted under climate change or on days of extreme fire danger when conditions are warmer, drier and windier. Under extreme conditions, weather is likely to be a greater predictor of fire size and spread than fuel load [58] so it can be expected that there will be instances

in which the weather will overwhelm the capacity of people to manipulate fire behaviour through fuel modification. Similarly, there will be conditions that render the outdoor environment too hot for human comfort. It should be noted that the PET was modelling used the Ray-Man default parameters of a healthy young male to demonstrate heat stress, not those of the demographics most vulnerable to the effects of heat.

Another future research opportunity would be to analyse the spatial arrangement of the open space scenarios in relation to other landscape elements such as buildings and roads. Shading from buildings [59], street orientation [10, 60] and ground surfaces [61] have all been found to influence cooling patterns of urban areas and correlations have been established between the size and location of urban greenspace and the cooling of the surrounding environment [4, 6, 62]. Landscape complexity also affects fire behaviour as buildings, roads and gardens all influence wind speeds at ground level [23]. Additionally, the proximity of houses to bushland has been associated to property loss [11].

Further considerations are the many other social and environmental values and trade-offs to consider, all worthy of further investigation in their own right. Whilst outside of the scope of this study, it should be recognised that the communities that reside in bushfire prone areas do so for a multitude of reasons [63–67] and personal preferences towards vegetation largely depend on the values of each individual [68]. Understanding these values will further help governments, urban planners, landscape architects, fire agencies and residents plan and manage safe, healthy and appealing urban environments.

Potentially then, the question should be not what vegetation is planted in urban landscapes but where and how human activities and settlements are planned in relation to them. A consideration of how different planting compositions burn may help determine how much of a fuel break is acceptable to ensure that the community can still benefit from the many critical services that urban vegetation provides.

## Conclusion

Understanding how fire and urban heat play out spatially is essential for the design and management of urban vegetation in fire prone areas to ensure that we are protecting citizens from both fire and heat, without compromising the additional aesthetic, biodiversity and other benefits associated with complex vegetation. This study provides preliminary evidence that demonstrates the potential for these potentially antagonistic goals to be considered together, and highlights the opportunity for maximising benefits while simultaneously reducing risks through decisions around the vertical and horizontal arrangements of plants in the landscape.

## Supporting information

**S1 File. Forest flammability model inputs.**
(PDF)

**S2 File. Rayman model inputs.**
(PDF)

## Acknowledgments

The research presented here was performed while TM was undertaking the subject FRST90077 Long Research Project B under the supervision of TDP and AKH as part of the Master Forest Ecosystem Science at The University of Melbourne. We would like to thank James MacLeod for assistance in preparing Fig 1 and Phil Zylstra for advice in the use of the Forest Flammability Model.

## Author Contributions

**Conceptualization:** Tania A. MacLeod, Amy K. Hahs, Trent D. Penman.

**Data curation:** Tania A. MacLeod.

**Formal analysis:** Tania A. MacLeod, Trent D. Penman.

**Investigation:** Tania A. MacLeod.

**Methodology:** Tania A. MacLeod, Amy K. Hahs, Trent D. Penman.

**Project administration:** Tania A. MacLeod, Trent D. Penman.

**Software:** Trent D. Penman.

**Supervision:** Amy K. Hahs, Trent D. Penman.

**Visualization:** Tania A. MacLeod, Trent D. Penman.

**Writing – original draft:** Tania A. MacLeod, Amy K. Hahs.

**Writing – review & editing:** Tania A. MacLeod, Amy K. Hahs, Trent D. Penman.

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
