## [Decision Letter · Decision Letter 0]

12 Sep 2019

PONE-D-19-21080

How can planting arrangement balance the risks of heat and fire on human settlements in a fire-prone urban landscape?

PLOS ONE

Dear Dr Hahs,

Thank you for submitting your manuscript to PLOS ONE. After careful consideration, we feel that it has merit but does not fully meet PLOS ONE’s publication criteria as it currently stands. Therefore, we invite you to submit a revised version of the manuscript that addresses the points raised during the review process.

We would appreciate receiving your revised manuscript by Oct 27 2019 11:59PM. To enhance the reproducibility of your results, we recommend that if applicable you deposit your laboratory protocols in protocols.io, where a protocol can be assigned its own identifier (DOI) such that it can be cited independently in the future. For instructions see: http://journals.plos.org/plosone/s/submission-guidelines#loc-laboratory-protocols

We look forward to receiving your revised manuscript.

Kind regards,

Monjur Mourshed, Ph.D., B.Arch.

Academic Editor

PLOS ONE

Journal Requirements:

'The authors have declared that no competing interests exist.' 

We note that one or more of the authors are employed by a commercial company: Hahs Consulting Pty Ltd.

Additional Editor Comments (if provided):

I have had an opportunity to read the manuscript and the reviewers' comments. Please consider the following points:

Generalisations: Like most models, the proposed one only considers a subset of the interacting factors, as pointed out by both reviewers. The authors are advised to discuss the limitations of their approach and revisit the generalisations made so that the contexts of the interpretations are clearer to the reader.

Consideration of the factors: Reviewer 2 points out that the availability of water in both soil and plant tissue is a factor that needs to be considered if the model is meant for wider application. The authors are advised to consider this point in the revised manuscript.

Reviewers' comments:

Reviewer's Responses to Questions

**Comments to the Author**

1. Is the manuscript technically sound, and do the data support the conclusions?

Reviewer #1: Yes

Reviewer #2: No

2. Has the statistical analysis been performed appropriately and rigorously? 

Reviewer #1: Yes

Reviewer #2: I Don't Know

3. Have the authors made all data underlying the findings in their manuscript fully available?

Reviewer #1: Yes

Reviewer #2: Yes

4. Is the manuscript presented in an intelligible fashion and written in standard English?

Reviewer #1: Yes

Reviewer #2: Yes

5. Review Comments to the Author

Reviewer #1: How can planting arrangement balance the risk of heat and fire on human settlements in a fire-prone urban landscape?

The study modeled planting arrangements in a hypothetical unbuilt environment in a bushfire prone region of Australia. The topic is of extreme importance, as the authors indicated in the implications, lines 373 – 384. Permit exceptions to remove large overstory trees or any vegetation within 50 meters of a structure in a bushfire management overlay completely ignores the impact this would have on human tolerance of high heat conditions. The unfortunate tradeoff of removing the possibility for fire by removing trees is obvious, but it hides a very serious (more serious?) threat (line 87) of heat-related exhaustion, fatigue, and death. The policy clearly is targeted to protect property over the life of vulnerable populations, infants, children with asthma, elderly, and those living in impoverished/downwind areas with higher levels of exposure to ambient air. In this light, the study would have been strengthened if it would have modeled a vulnerable human indicator, rather than a young, healthy male wearing appropriate clothing.

Additional Comments:

The short title “Balancing fire risk and human thermal comfort in urban landscapes” is a closer match to the article than the long title. The long title seems to suggest that the paper would look at multiple planting arrangements, however it largely emphasized one particular model where the fuel of shrubs and trees overlapped. This led to the overgeneralized suggestion that trees would be better off without any vegetation underneath, when looking at fire risk, and as the authors note, turning a blind eye to the ecology of urban environments. Figure 1 should correctly show what was modeled, so readers have a clearer understanding that it isn’t the presence/absence that played the most critical role, but the tight relationship between interconnected shrubs and trees.

Lines 61 – 68 place the paper within the established context of urban heat island effect. The models never really dive into UHI, since it is a hypothetical undeveloped park space. Further, the strength of the paper is in noting the unhealthy heat conditions in cities following climate change and urbanization. How do we balance human health and fire risk? This is a critical question that is not clearly emphasized in UHI models. The fact that policy aims to make people in cities more vulnerable to heat is a much greater threat at a public health level than UHI. It would be great if the authors could emphasize the policy implications sooner, as places like California are looking for heat/fire balance more than UHI.

Evapotranspiration is mentioned but, I’m not sure what the context is, does this matter in an seasonal context with low humidity and low precipitation? It seems like a tangent that is never really tested. Might be a good idea to remove it.

Overall, excellent article and on point with urban conditions around the world facing a very credible threat of protecting high property values over human health by removing the fuel. Comments above are recommendations to improve the article. Publish soon!

Reviewer #2: This is a well-written manuscript that seeks to develop fire-wise guidelines for planting arrangements in managed landscapes. Unfortunately, the most important factor - water availability in both the soil and plant tissue - is not considered. While this model might work well in production forest management (where irrigation and other types of routine landscape management are not practical), it does not translate well to managed landscapes, which by their very definition are managed, sometimes intensively. More inclusion and discussion of literature relevant to managing urban landscapes is needed. This will help to highlight the important environmental factors at play in flammability - not just foliar traits and plant placement.

In a well-managed, diverse landscape, you'll have a variety of trees, shrubs, groundcovers, and so on, which are optimally maintained by a combination of thick, woody mulches and sufficient water input. Wood chip mulches not only cool the soil and reduce evaporation, they store water and therefore do not dry out, unlike bark mulches and other more flammable materials. These landscapes are highly resistant to fire, because the soil is moist, the mulch is not flammable, and the vegetation well hydrated. A closed canopy will create an underlying landscape that is cooler, moister, and the most resistant to fire. We know this to be true through decades of observing relatively intact natural forests in moister climates. It's the open canopies, with isolated trees, that are at risk because the exposed, open landscape between the trees dry out and be more likely to burn.

Any practcial recommendations in this manuscript need to be removed, because the model under study does not contain the crucial environmental factors needed to make it relevant to well-managed landscapes. Plant-soil-water relations are crucial in this discussion, as the authors acknowledge in line 415.

6. PLOS authors have the option to publish the peer review history of their article (what does this mean?). If published, this will include your full peer review and any attached files.

Reviewer #1: Yes: Benjamin A. Shirtcliff, Ph.D.

Reviewer #2: Yes: Linda Chalker-Scott

---

## [Author Response · Author response to Decision Letter 0]

3 Nov 2019

- Generalizations. Greater discussion around the model and its limitations has been included in the discussion and text has been amended in a number of locations to clarify the assumptions made from our results. 

-Consideration of the factors. The discussion about fuel moisture content and the wider applications of the Forest Flammability Model has been expanded in lines 416 to 430 to address Reviewer 2’s comments. 

Responses to Reviewer 1’s comments:

• UHI has been omitted from the introduction and framing of the article and instead addressed as urban heat

• The seriousness of urban heat as a killer has been emphasized on lines 88-89. 

• We have stated that the study would have been further strengthened with a vulnerable cohort being used for the human climatology data in line 482.

• The long title has been changed to “Balancing fire risk and human thermal comfort in fire-prone urban landscapes”

• Policy implications are introduced earlier in the article in line 80. 

• Given Reviewer 2’s comments with regard to moisture availability, the discussion on evapotranspiration was not omitted from the article but addressed in a discussion on plant traits and moisture availability lines 416-430. 

Thank you for your constructive comments.

 Responses to Reviewer 2’s comments:

• In response to Reviewer 2’s comments, water availability and plant-soil relations are discussed further in 416-430.

• There seemed to be some confusion over the type of landscapes that were modeled and the application of the Forest Flammability Model. As such, the methods section now includes a sentence of the typical open space types that were observed and modeled. This may help readers from outside the study area identify with the common landscape types and vegetation themes that were modeled and how the FFM was applied. These landscapes typically were not intensively managed, mulched or irrigated landscapes as Reviewer 2 suggests. Refer to lines 128-132. 

• The FFM was not intended for production forestry as suggested by Reviewer 2, but for measuring flammability of forests based on the flammability of individual fuel elements including ladder and surface fuels as well as environmental and climatic variations such as slope, wind-speed, relative humidity and temperature. 

• As with reviewer #1, generalizations, limitations and practical recommendations in regard to the interpretation of results have been better phrased within the discussion. 

Many thanks for your feedback.

---

## [Decision Letter · Decision Letter 1]

18 Nov 2019

Balancing fire risk and human thermal comfort in fire-prone urban landscapes

PONE-D-19-21080R1

Dear Dr. MacLeod,

We are pleased to inform you that your manuscript has been judged scientifically suitable for publication and will be formally accepted for publication once it complies with all outstanding technical requirements.

With kind regards,

Monjur Mourshed, Ph.D., B.Arch.

Academic Editor

PLOS ONE

Additional Editor Comments (optional):

Reviewers' comments:

Reviewer's Responses to Questions

**Comments to the Author**

1. If the authors have adequately addressed your comments raised in a previous round of review and you feel that this manuscript is now acceptable for publication, you may indicate that here to bypass the “Comments to the Author” section, enter your conflict of interest statement in the “Confidential to Editor” section, and submit your "Accept" recommendation.

Reviewer #1: All comments have been addressed

Reviewer #2: All comments have been addressed

2. Is the manuscript technically sound, and do the data support the conclusions?

Reviewer #1: Yes

Reviewer #2: Yes

3. Has the statistical analysis been performed appropriately and rigorously? 

Reviewer #1: Yes

Reviewer #2: I Don't Know

4. Have the authors made all data underlying the findings in their manuscript fully available?

Reviewer #1: (No Response)

Reviewer #2: Yes

5. Is the manuscript presented in an intelligible fashion and written in standard English?

Reviewer #1: (No Response)

Reviewer #2: Yes

6. Review Comments to the Author

Reviewer #1: (No Response)

Reviewer #2: Thank you for addressing my concerns with additional information and discussion. My "I don't know" response to the statistical analyses question is due to my lack of familiarity with running stats on computer models. I am assuming other reviewers will be more aware of these details.

7. PLOS authors have the option to publish the peer review history of their article (what does this mean?). If published, this will include your full peer review and any attached files.

Reviewer #1: Yes: Benjamin Shirtcliff

Reviewer #2: Yes: Linda Chalker-Scott

---

## [Editor Report · Acceptance letter]

10 Dec 2019

PONE-D-19-21080R1 

Balancing fire risk and human thermal comfort in fire-prone urban landscapes 

Dear Dr. MacLeod:

I am pleased to inform you that your manuscript has been deemed suitable for publication in PLOS ONE. Congratulations! Your manuscript is now with our production department. 

With kind regards,

on behalf of

Prof. Monjur Mourshed 

Academic Editor

PLOS ONE